# Parcellation-Based Connectivity Model of the Judgement Core

**DOI:** 10.3390/jpm13091384

**Published:** 2023-09-16

**Authors:** Jorge Hormovas, Nicholas B. Dadario, Si Jie Tang, Peter Nicholas, Vukshitha Dhanaraj, Isabella Young, Stephane Doyen, Michael E. Sughrue

**Affiliations:** 1Centre for Minimally Invasive Neurosurgery, Prince of Wales Private Hospital, Level 7 Prince of Wales Private Hospital, Randwick, NSW 2031, Australia; jorgehormova@gmail.com (J.H.); vukshi27@gmail.com (V.D.); 2Robert Wood Johnson Medical School, Rutgers University, 125 Paterson St., New Brunswick, NJ 08901, USA; nicholas.dadario@rutgers.edu; 3School of Medicine, 21772 University of California Davis Medical Center, 2315 Stockton Blvd., Sacramento, CA 95817, USA; 4Omniscient Neurotechnology, Level 10/580 George Street, Haymarket, NSW 2000, Australia; peter.nicholas@o8t.com (P.N.); isabella.young@o8t.com (I.Y.); stephane.doyen@o8t.com (S.D.)

**Keywords:** judgement, networks, fMRI, meta-analysis, brain connectome, neurosurgery

## Abstract

Judgement is a higher-order brain function utilized in the evaluation process of problem solving. However, heterogeneity in the task methodology based on the many definitions of judgement and its expansive and nuanced applications have prevented the identification of a unified cortical model at a level of granularity necessary for clinical translation. Forty-six task-based fMRI studies were used to generate activation-likelihood estimations (ALE) across moral, social, risky, and interpersonal judgement paradigms. Cortical parcellations overlapping these ALEs were used to delineate patterns in neurocognitive network engagement for the four judgement tasks. Moral judgement involved the bilateral superior frontal gyri, right temporal gyri, and left parietal lobe. Social judgement demonstrated a left-dominant frontoparietal network with engagement of right-sided temporal limbic regions. Moral and social judgement tasks evoked mutual engagement of the bilateral DMN. Both interpersonal and risk judgement were shown to involve a right-sided frontoparietal network with accompanying engagement of the left insular cortex, converging at the right-sided CEN. Cortical activation in normophysiological judgement function followed two separable patterns involving the large-scale neurocognitive networks. Specifically, the DMN was found to subserve judgement centered around social inferences and moral cognition, while the CEN subserved tasks involving probabilistic reasoning, risk estimation, and strategic contemplation.

## 1. Introduction

Judgement is a complicated aspect of higher cognition that relates to the capacity to make decisions and form reasonable opinions [1]. Despite its ubiquitous description in the literature as a distinguishing achievement of human neurology, judgement as a cognitive faculty, along with its underlying network architecture, remains more or less loosely defined. Consequently, efforts to preserve one’s capacity for judgement have proved a lackluster endeavor in the field of neurosurgery.

Terms including judgement, decision making, problem solving, and reasoning are often used interchangeably throughout the neuropsychiatric literature [2,3]. Judgement should be understood as the evaluative process encountered after the stage of active problem solving [4,5]. While problem solving involves the integration of information to provide an individual with strategies to respond to the task/problem at hand, judgement relates to the relative appraisal of these options to select the one most fitting [1]. A lack of consistent definitions and clear neuroanatomic models hinders the effectiveness of targeted clinical management, with one report showing that across a population of clinicians, a total of 185 different tools were used to test judgement [2]. As such, creating an anatomic model of neurological judgement function that is based on a clearly specified definition in the literature is of immense clinical significance.

Increasing evidence from the field of “connectomics” has more accurately suggested that the brain involves dynamically interacting brain networks that are efficiently organized to subserve complex human thinking and behavior [6]. Of particular interest for the current study are regions that are said to be “functionally connected”. Two spatially distant regions are said to be functionally connected when they demonstrate highly synchronized activity, often measured by changes in blood-oxygenation-level-dependent (BOLD) signals, within a closely aligned temporal window and for similar tasks [7]. Importantly, attempting to create a connectomics-based neuroanatomic model of judgement function will likely better accommodate for its expectedly distributed neuroarchitecture than would be made possible by the traditional localizationist perspective [8].

The present study aims to develop a clearer picture of exactly where in the brain the faculty of judgement is localized, particularly in the areas of moral judgement, self/other-referential judgement, and judgement in conditions of variable certainty. To address this, we conducted an activation-likelihood estimation (ALE) coordinated-based meta-analysis (CBMA), using data from task-based functional magnetic resonance imaging (fMRI) studies of healthy participants. Our study integrated data from a range of experimental paradigms, grossly divided into four subgroups on the basis of the task modality used within each. In this way, we aimed to reduce the manifold expressions of judgement into a unified pattern of network engagement and, furthermore, to differentiate between brain areas that serve computational roles essential for all judgement and accessory regions preferentially engaged for certain applications. Furthermore, our results are reported in an anatomically specific parcellation nomenclature, which may allow our preliminary model to be utilized as a basis for future study.

## 2. Method

### 2.1. Literature Retrieval and Study Categorisation

A literature search was conducted across two databases (Pubmed and Embase) using the search queries “Judgement & fMRI” on Pubmed and “(Judgement OR decision-making) AND (fMRI OR functional magnetic resonance imaging OR BOLD signal)” on Embase. The literature was screened in accordance with PRISMA guidelines [9,10] and current best practice guidelines for CBMA70 [11].

The primary selection of articles was directed by the main screening criteria outlined in Figure 1. Peer-reviewed studies were included if they included healthy human subjects age 18 to 65. Studies were required to express results with Talairach or Montreal Neurological Institute (MNI) standardized coordinates based on whole brain, voxel-wise imaging, and reported foci with a voxel-level threshold of uncorrected *p* value < 0.001 or a corrected cluster probability of *p* value < 0.05. Studies were required to have task-based fRMI with an active judgement task of either making a decision or an evaluation. A matched or baseline control task was also required to be performed. Studies were required to have eight or more study participants. All studies met current best practice guidelines for CBMA, except for one study that had a total of eight study participants.

### 2.2. Activation Likelihood Estimation

All Talairach coordinates were converted to the MNI stereotaxic space using an online conversion tool available at https://bioimagesuiteweb.github.io/webapp/mni2tal.html (accessed on 1 October 2021). The BrainMap GingerALE 3.0.2 was then used to generate an ALE for each sub-analysis at a cluster level threshold of 0.05, threshold permutation of 1000, and uncorrected *p* value less than 0.001 [12,13].

The activation likelihood estimation treated each (x,y,z) coordinate as a spatial uncertainty distribution modelled by a three-dimensional gaussian kernel, the center of which was equivalent to the location of the peak cortical activation encoded in the BOLD-contrast [14]. The radius of each kernel is equivalent to the uncertainty associated with those particular foci. This uncertainty is a function of the innate limitations of the study design and techniques, a core variable being the number of study participants for example [12]. Ultimately, a voxel-wise integration of activation probabilities across all experiments was conducted to generate a modeled activation map. This map was compared with a null-distribution that assumed there was no spatial association between the reported foci. Voxels that demonstrated a likelihood for convergence above the aforementioned statistical thresholds remained in the final ALE products [14]. These ALE coordinate data were displayed on an MNI-normalized brain template using the Multi-image Analysis GUI (Mango) 4.0.1 (ric.uthscsa.edu/mango, 10 December 2021).

### 2.3. Matching of ALE Clusters to HCP Parcellations

ALE clusters were then matched for volumetric overlap with the cortical parcellations described under the human connectome projects (HCP) multimodal parcellation scheme. A complete table of correspondences describing the MNI locus of each ALE cluster, intersected HCP parcellations and the percentage overlap between them is recorded in Appendix A.

### 2.4. Matching of Parcellations to Functional Networks

We then matched all identified parcellations to their belonging as a component of one of the core brain networks as described under Yeo’s 7 network model [15]. These include the salience, dorsal attention, limbic, central executive (CEN), default mode (DMN), visual, and sensorimotor networks. The correspondences between parcellations and neurocognitive networks are shown in Appendix A.

### 2.5. Contrast Analysis

ALE contrast analysis was conducted to identify potential core and accessory brain regions involved in judgement. A core region is defined as a brain area that is found to be activated across multiple ALEs, whereas an accessory region is found to be uniquely associated with a given modality or paradigm subtype. Patterns of large-scale network engagement were uncovered by relating these brain regions to the broader networks they formed a component of.

### 2.6. BrainMap Behavioral Analysis

To examine how our ALE coordinate data for each judgement paradigm were related to behavioral experiments in the literature, a behavioral analysis across moral, social, risky, and interpersonal judgement paradigms was completed. The ALE coordinate data were analyzed using the activation coordinate experiment-wise search (ACES) on BrainMap (https://www.brainmap.org/, accessed on 1 January 2022) to find the top most similar behavioral experiments (based on task-related activations) in the BrainMap database to each judgement paradigm. The data from the top 10 experiments from each judgement paradigm were retrieved (the BrainMap ID, the experiment size, the number of coordinates matching our ALE data, and the coordinate similarity percent).

These data are presented in Appendix A.

## 3. Results

Fifty-six viable datasets were identified and organized into four modality-based sub-analyses. These subgroups were designed to maximize the number of studies we could include in the final analysis while optimizing the paradigm homogeneity within any one grouping. Additionally, sub-analyses were defined in accordance with the modality-defined domains described in Supplement 1 so that this study’s vernacular was not only consistent with prior research, but the conclusions could be better interpreted in the context of prior findings. The sub-analyses were defined as follows: social judgement (SJ), which involves making a judgement of another person based on incomplete information; moral judgement (MJ), which involves making a judgement between two opposing scenarios or moral principles; risk judgement (RJ), which involves those tasks where decision outcomes can cause potential financial or physical harm to exclusively the decision maker themselves; and interpersonal judgement (IJ), which includes paradigms where choice outcome is influenced by the choice of another person. Unfortunately, 10 studies that met our inclusion/exclusion criteria were excluded because they could not be delegated to an analysis without compromising the homogeneity of the dataset. A complete description of the included studies is listed in Appendix A.

### 3.1. Judgement Activation Maps

Figure 2 demonstrates the ALE of the 10-study meta-analysis completed for the MJ subgroup. This showed a multiregional functional network anchored between the bilateral superior frontal gyri (SFG), right superior and middle temporal gyri, and left parietal lobe (Figure 2a–d). A comparative analysis between the four primary ALE clusters and the HCP atlas ROIs is illustrated in Figure 2e. This revealed the involvement of 10 left-sided parcellations including areas 9m and d32 in the SFG; areas PGi, TPOJ2, STV, 31pv, and 31pd in the inferior parietal lobe (IPL); 7m in the precuneus; and d23ab and v23ab in the posterior cingulate cortex (PCC). Similarly, eight parcellations were identified in the right hemisphere, including areas 9m and d32 in the SFG; 7m in the precuneus; and areas TE1a, STGa, STSva, STSda, and TGd in the temporal lobe.

Figure 3 demonstrates the ALE of the 17-study meta-analysis conducted for the SJ subgroup. This included a left-dominant frontoparietal network with concomitant engagement of right-sided temporal limbic regions (Figure 3a–d). A comparative analysis between the three primary ALE clusters and the HCP atlas ROIs is illustrated in Figure 3e. This revealed functional involvement of 10 left-sided parcellations including areas 47s, 47I, 47m, 45, 8BM, and SCEF in the frontal lobe; SFL in the supplementary motor area; AVI in the insular cortex; and FOP4 in the frontoparietal operculum. Similarly, five right-sided parcellations were identified, including areas pOFC and 8BM in the frontal cortex, Pir in the insula, and EC and the amygdala in the temporal lobe.

Figure 4 demonstrates the ALE of the 11-study meta-analysis conducted for RJ. This revealed a right-sided frontoparietal network with accompanying engagement of the left insular cortex (Figure 4a–d). A comparative analysis between the four primary ALE clusters and the HCP atlas ROIs is shown in Figure 4e. This revealed functional involvement of 11 right-sided parcellations, including areas PFm, IP2, AIP, and LIPd in the IPL; p32pr and a24pr in the anterior cingulate; and a32pr, 8BM, p9-46v, IFSp, and 46 in the frontal lobe. Additionally, areas MI and AAIC were identified in the left-sided insula.

Figure 5 demonstrates the ALE of the eight-study meta-analysis conducted for the IJ subgroup. This highlighted a right-sided frontoparietal network with accompanying engagement of the bilateral insular cortices (Figure 5a–d). A comparative analysis between the four primary ALE clusters and the HCP atlas ROIs is illustrated in Figure 5e. This revealed functional involvement of 17 right-sided parcellations, including areas p32pr, a24pr, p24, and 33pr in the cingulate cortex; 8BM, SCEF, a32pr, and d32 in the frontal lobe; MIP, LIPd, IP1, IP2, FOP5, and FOP4 in the parietal lobe; and MI, AVI, and AAIC in the insula. Additionally, activation of parcellation MI in the insula and FOP4 in the frontoparietal operculum was shown for the left hemisphere.

### 3.2. Parcellation-Based Comparative Analysis

Demonstrated in Figure 6, cortical co-activation between risk and interpersonal judgement was observed within the left insula and a range of frontoparietal parcellations belonging principally to the right-sided central executive and salience networks. Moral and social judgement demonstrated no parcellation overlap but, however, evoked a strong mutual activation of the bilateral DMN.

### 3.3. Judgement Literature Behavioral Analysis

A behavioral analysis across moral, social, risky, and interpersonal judgement paradigms was also completed. ALE coordinate data, consisting of anatomical coordinates most representative of each judgement paradigm, were analyzed using the activation coordinate experiment-wise search (ACES) on BrainMap to find the top most similar behavioral experiments (based on task-related activations) to each of these judgement paradigm coordinates in the BrainMap database. These data are presented in Appendix A.

Broadly, each paradigm was related to experiments with activations including similar aspects of judgement. Notably, experiments were most similar between moral and social judgements, while risk and interpersonal judgements were more similar. Risk and interpersonal judgement paradigms demonstrated similar activations in studies that examined stimuli in the environment and between others generally for the purpose of a reward or calculated decision making. Risk judgement had activation coordinate similarities to studies that examined adverse stimuli (e.g., painful facial expressions, negative pictures, and trauma) and performance feedback. Interpersonal judgement had coordinate similarities shared across studies more related to interactions between oneself and other subjects or objects in the environment, such as transitive inference (inferring relations between objects), pattern matching, decisions according to reward probability, and provocation in a situation. When examining moral judgement, coordinate similarities from our moral activation map included a number of studies examining opposing scenarios, such as meaningful versus non-meaningful stimuli, two different body movements, and rest versus action. Social judgement had coordinate similarities to the social activation map across studies that examined a variety of emotional stimuli, such as emotional pictures, depression, and also responses to painful stimuli or scenarios.

## 4. Discussion

Our study found that cortical activation in normophysiological judgement function follows two general and separable patterns involving the large-scale neurocognitive networks. Specifically, the DMN was found to subserve judgement centered around social inferences and moral cognition, while the central executive network (CEN) subserved tasks involving probabilistic reasoning, risk estimation, and strategic contemplation. Consequently, we conjecture that the average age of competency observed across different decision-making modalities [16] is actually reflective of the distinct developmental time-courses required for maturity of the corresponding networks. To arrive at these conclusions, we conducted an ALE CBMA across four distinct task types including moral, social, risky, and interpersonal judgement paradigms. By determining regions of cortical activation shared across heterogenous decision-making paradigms, we are more capable of delineating the computational significance of these identified regions for judgement as a whole. Additionally, by characterizing the broader functional circuitry that subserves judgement, we offer a more clinically actionable framework for understanding related cognitive and affective dysfunction in neuropsychiatric pathology [17].

The functional reliance of judgement on the DMN, SN, and CEN aligns with recent work that has attempted to systematically resolve the diverse phenotypic expression of psychopathology to the aberrant connectivity and organization of these same neurocognitive networks [18]. Referred to as the “triple network theory”, it has been demonstrated that the etiology and symptomatology of prevalent conditions like schizophrenia [19], depression [20], or Alzheimer’s disease [21] can be largely explained by patterns of dysregulation across these three intrinsic connectivity networks (ICNs). For example, it has been shown that impaired social decision making in autism spectrum disorder (ASD) is related to hypoactivation of the salience network, particularly its right anterior insula (AI) node [22]. Historically, this has been interpreted as driving the inability for ASD patients to attach appropriate importance to socially salient inputs, observed clinically as a lack of eye contact and/or verbal responsiveness [18]. Given the strong evidence that the right AI is particularly critical to the dynamic recruitment of other large-scale brain networks [23], in the context of our results, we can suspect that the clinical features of ASD are additionally attributable to improper recruitment of the temporal DMN. Ultimately, the fact that healthy judgement function relies on the same triple network system implicated in a range of the most prevalent neuropsychiatric conditions offers a mechanistic connection as to why decision-making deficits are observed across so many of these same conditions.

### 4.1. Network Evidence for Two Judgement Systems

Neurocognitive network engagement for judgement was found to robustly separate into two systems: a DMN system implicated for moral and social tasks and a salience/CEN system implicated for risky and interpersonal tasks. Furthermore, the DMN system was found to be left dominant, while the salience/CEN system was strongly right dominant.

### 4.2. DMN Judgement

Despite an absence of parcellation co-activity between moral and social judgement, there was strong shared activation of the DMN between these sub-analyses. Conceptually, the MJ and SJ active tasks were unified in their dependence on inferring the “goodness” of a target stimulus and integrating this into a behavioral decision. While the DMN was first discovered as a resting-state ICN [24], our work adds to an emerging body of evidence that further characterizes the DMN as an essential mediator of a number of active processes including moral cognition [25], theory of mind [26], empathy [27], self-referential contemplation [28], and social behavior [27]. Consistent with this assessment, Jung et al. demonstrated that reduced inter-network connectivity between the amygdala (of the limbic network) and the ventromedial PFC (a DMN node) is correlated with reduced moral competency in otherwise neuropsychiatrically healthy adults [29]. Similarly, a linear anticorrelation between social dysfunction and intra-network DMN connectivity was observed across a population of major depressive disorder patients [20]. In the same way, the propensity for harmful (or utilitarian) choices in moral dilemma paradigms was correlated with reduced anteroposterior integration of the DMN in a sample of incarcerated psychopaths [30]. Despite these alignments, a network-based study by Chiong et al. demonstrated that by framing MJ paradigms in a third-person context, there was a stronger pattern of CEN engagement and DMN disengagement than what would be expected [31]. Ultimately, although the literature and our findings do not unequivocally establish the DMN as the sole mediator of these processes, we do suggest that the DMN is the most essential driver of this class of socio-inferential judgement.

### 4.3. CEN Judgement

Judgement under conditions of risk and interpersonal judgement evoked strong co-activation of the right-sided CEN and bilateral SN. Conceptually, the experimental tasks included within these two sub-analyses were similar in that they involved the pursuit of reward maximization, outcome anticipation, and strategizing. While the role of the CEN in goal-directed judgement [18] and the manipulation of information in working memory [32] has been demonstrated elsewhere, we have shown it is precisely the right-sided parcellations of this structurally bilateral network [33] that mediate these functions in the context of judgement. Of the right-sided CEN parcellations, this pairing evoked mutual overlap in R_a32pr, which has in prior studies been shown to mediate outcome anticipation and recognition of reward values [34,35]. Similarly, there was a high degree of overlap in the R_IP2 parcellation previously implicated in mathematical processing [36], congruent with the probabilistic reasoning involved in these judgement types. There was also strong overlap across RJ and IJ in the left-sided insula, precisely of its L_MI parcellation located at the posterosuperior aspect of the short insular gyrus. Pointedly, the insula is a complicated area and has been divided into as many as 13 discrete functional territories [37], including regions involved in judgement under conditions of risk and uncertainty [38]. Additionally, patients with insula lesions were demonstrated to become less sensitive to differences in expected value between judgement alternatives, producing recognizably suboptimal decision-making behaviors [39]. Ultimately, whether these deficits were the result of a loss of directly contributing eloquent tissue located in the insula itself or due to the functional decoupling of CEN activity by proxy of that same insula damage is yet to be resolved.

### 4.4. A Neurodevelopmental Perspective of the Two Judgement Systems

By age two, the cerebral network topology has matured such that the salience, default mode, and central executive networks are recognizable [40]. From this point forward, these networks follow patterns of global architectural restructuring universally hallmarked by a reduction in short-ranged association fibers and a strengthening of long-range projection fibers [41]. Despite understanding of these global trends, the precise neurodevelopmental trajectories of individual networks remains an area of active research [42]. During ages 7–15, the DMN undergoes critical growth [43] with strengthening of white matter connections between the PCC and medial PFC observed as a cardinal development [44] on its course to complete integration by age 21 [45]. Intriguingly, this period of DMN development is concomitant with the intense period of social reorientation and maturation that occurs during adolescence [46]. The synchronized time course of DMN and socio-behavioral maturation, together with our finding of the shared evocation of the DMN and its PCC/medial PFC nodes across moral and social judgement tasks, reinforces our suggestion that the DMN is a core guide of social behavior.

We have also shown that judgement involving strategic thinking, planning, and risk estimation is largely subserved by the CEN. Central to CEN functionality is its dynamic relationship with the SN, largely mediated by connections through the AI [47]. Lebel and Beaulieu found that of all long-range axonal bundles, the fronto-occipital fasciculus (which mediates the connection between the SNs AI and CENs dorsolateral PFC [44]) hosted the greatest increase in connectivity strength over a 1–6 year interval in a young adult population [48]. Mirroring these late-occurring structural developments, performance across these same executive functions was shown to not reach a stable maximum until approximately age 30 [49]. Subsequently, we conjecture that the CEN reaches maturity significantly later than the DMN on an order of about 10 years. Furthermore, we suspect that the delayed achievement of full adult competency in goal-directed decision-making tasks is in part attributable to the longer maturational timeline required for the full development of the inter-network connectivity between the SN and CEN.

### 4.5. Significance of Findings and Future Directions

#### Neurosurgical Conservation of Judgement Function

One of the core exercises of neurosurgery is the optimization of onco-functional balance, that is, to maximize the extent of tumor resection while minimizing the induction of new neurological deficits [50,51]. Historically, this has been achieved by experiential familiarity with eloquent zones on behalf of the surgeon. Additionally, the application of multimodal imaging techniques has further optimized surgical decision making, a notable example being the application of diffusion-tensor imaging to preoperative MRI, to indicate the use of non-conventional surgical corridors in cases where mass-occupying lesions have displaced or involved white matter tracts [52,53]. Furthermore, to account for interindividual variability in brain organization, awake intraoperative electrical stimulation has emerged as the gold standard for identifying eloquent tissue on a patient-by-patient basis [54]. Despite success in elucidating and therefore preserving certain visual, motor, and language areas, awake electrostimulation has proven to be poorly effective in monitoring the more complex and operationally subtle executive functions [55]. Ultimately, a lack of understanding for the neuroanatomical correlates of higher cognition, combined with an inadequacy for the traditional tools of functional conservation to have translated efficacy, provides a reason as to why brain tumor patients continue to show poor cognitive morbidity in higher-order brain functions like judgement.

Despite these limitations, some neurosurgeons have begun utilizing graph theory metrics as a non-specific method for conserving higher cognition by optimizing the brains post-operative capacity for informational integration [50]. One such measure is global efficiency, which is equivalent to the average inverse of the shortest distance between any two nodes in a brain network [55]. The optimization of global efficiency therefore implies that network cores (which serve as hyper-connected hub regions) should be left intact whenever possible [56]. Contrary to these indications, we have shown judgement to be operationally centralized around peripheral network areas, specifically the temporal DMN and the right-sided lateral components of the CEN. Subsequently, the preservation of these areas is indicated to be more essential for decision making than the cores themselves. Thus, we highlight that precise anatomical models, when used in conjunction with non-specific metrics like global efficiency, will likely improve neurosurgical outcomes as compared with indiscriminately relying on graph theory metrics alone.

### 4.6. Targeted Therapy for the Restoration of Judgement Function

Repetitive transcranial magnetic stimulation (rTMS) involves the application of extracranial magnetic fluctuations to induce electrical currents within the cortex [57]. This non-invasively prompts cortical reorganization by the targeted up-/down-regulation of brain networks [58] alongside the induction of transient neuro-hyperplasticity that accelerates the realization of extra-modal therapeutic outcomes [59]. Provided that focal brain lesions can cause widespread functional and structural deficits due to the potentiation of dyssynchronous neuronal activity [60], rTMS’s capacity to reinstate such network synchrony renders it a promising frontier for neurorehabilitation [50]. For example, by reducing the within-network functional connectivity of the DMN between its dorsolateral PFC and medial PFC nodes, rTMS has proven to be effective for alleviating the symptoms of treatment-resistant major depressive disorder [61]. Recently, rTMS has been developed to modulate multiple cortical targets through the use of a machine-learning algorithm to detect anomalies in connectomics based on Glasser’s multimodal parcellations [62,63]. We suspect that an improved model of the functional neuroanatomy of judgement will translate into a similar ability to target decision-making deficits in future.

### 4.7. Study Limitations

All ALE meta-analyses are inherently limited by the large number of studies required for inclusion to detect smaller effects and to negate the over-representation of any one dataset [11]. Of the four sub-analyses included in this study, only the SJ ALE met the current recommended threshold of 17–20 studies [64]. Conversely, the interpersonal, moral, and risk subcategories had only 8, 10, and 11 datasets respectively, and therefore, their ALE products were more susceptible to being skewed toward a smaller selection of individual papers. Despite this lack of power, all included analyses met the threshold of eight studies, which has been demonstrated to ensure that no more than 50% of the final ALE is attributable to the most dominant experiment [64].

There is also a potential for selection bias to skew our results. One source of this is the lack of deliberate balancing of paradigm subtypes within individual ALEs. As a result, the ALE products may over-represent some paradigm-specific evocations for task types that dominate the dataset. An additional confounder of selection bias is the potential for our results to overly represent a certain patient demographic. Task-based fMRI activations have been shown to often follow patterned deviations depending on the characteristics of the participant demographic, including the level of education [65], gender [66], spoken language [67], ethnicity [68], and sociocultural background [69,70]. The ALE meta-analysis is therefore susceptible to unknowingly globalizing demographic-specific cortical activations as being a ubiquitous statistical trend.

To eliminate spurious cognitive processing that is functionally associated yet non-essential to judgement, we selected a majority of studies that used a matched control task to produce their BOLD signal contrast. These matched tasks worked to eliminate accessory cortical activation inspired by visual, motor, and semantic cues featured in the active task [71]. Despite this advantage, these BOLD contrasts were not effective in eliminating more intricate parallel processing elements like metacognitive monitoring of decision confidence. Consequently, this could imply computational significance for regions not actually essential to judgement function due to consistent coactivation across experiments.

Finally, an unavoidable source of bias in this study is the subjective categorization of the ALE sub-analysis groups. These categories form the interpretative framework for this study, and so alternative categorizations could potentially show a different pattern of network engagement. While we sampled and examined four large components of judgement, the sum of these groups does not represent the function in its entirety. Consequently, there remain other task modalities that are often used to study judgement not included in our final analysis. Thus, whether these other domains of judgement organize under the two-system framework described above remains an area for future research.

## 5. Conclusions

Judgement is a complicated aspect of higher cognition that has been studied in a multitude of ways. Despite an abundance of research, inconsistency in the way judgement is defined and in the parcellation and network atlases used to guide the interpretation of these studies has limited the clinical actionability of past findings. Here, we have used ALE CBMA to demonstrate that judgement function broadly involves the separable engagement of the DMN and CEN. Dividing judgement paradigms into moral, social, interpersonal, and risk, we found that the bilateral superior frontal gyri, right temporal gyri, and left parietal lobe were implicated in moral judgement. The left-dominant frontoparietal network and right-sided temporal limbic regions were implicated in social judgement. Both interpersonal and risk judgement involved the right-sided frontoparietal network and left insular cortex. Although these findings are consistent with the functional imaging literature, some ALEs were under-powered, and so this preliminary model requires further validation. Nevertheless, we suspect that improved anatomical models of judgement such as described here will contribute to the improved preservation and restoration of judgement-related cognition in clinical practice in the future.

## Figures and Tables

**Figure 1 jpm-13-01384-f001:**
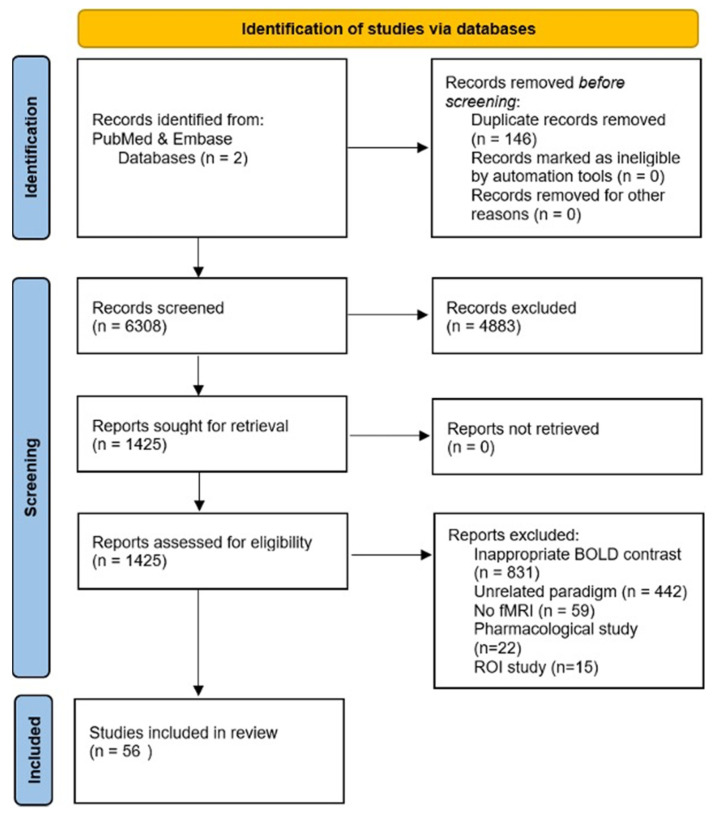
Summary of the literature review process (adapted from PRISMA guidelines) [9].

**Figure 2 jpm-13-01384-f002:**
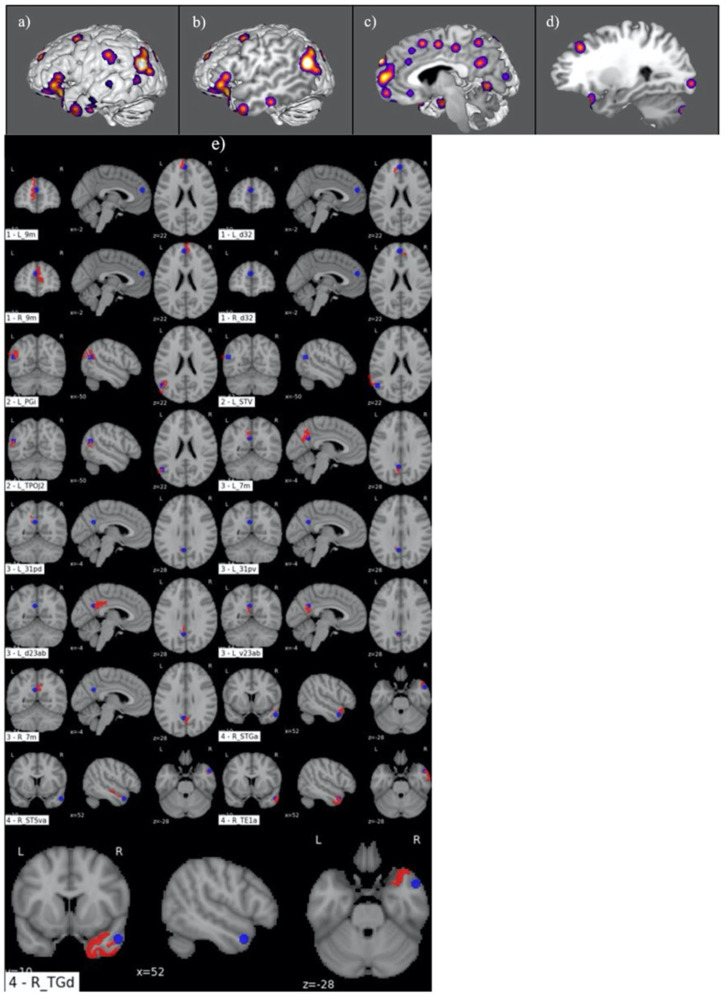
Graphical representation of ALE for moral judgement. (**a**–**d**) Illustration of ALE clusters represented on iterated sagittal slices of an MNI normalized brain template. (**e**) Volumetric overlap of ALE clusters with HCP parcellation territories.

**Figure 3 jpm-13-01384-f003:**
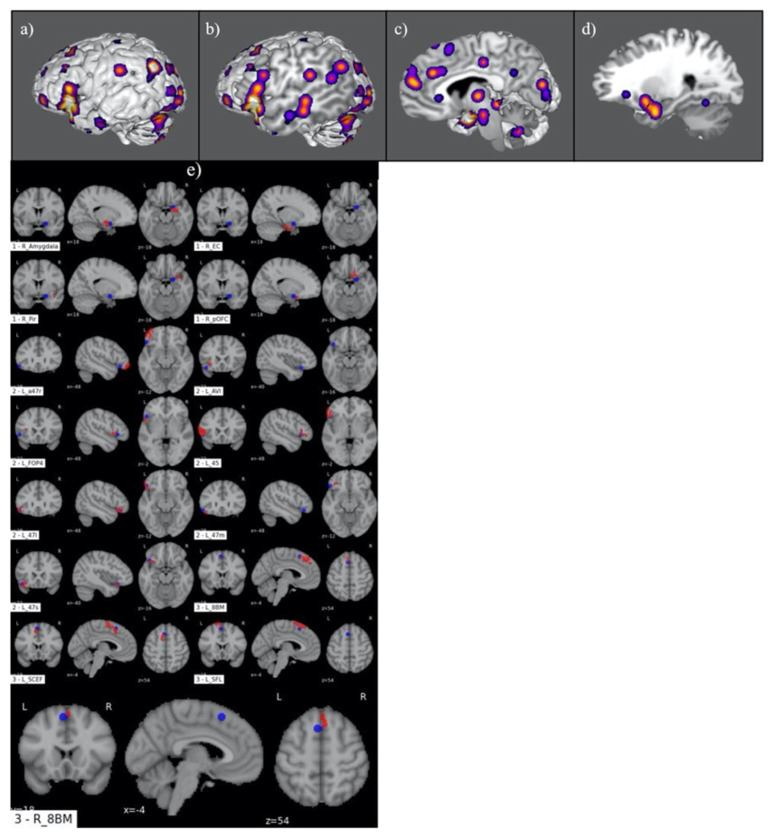
Graphical representation of ALE for social judgement. (**a**–**d**) Illustration of ALE clusters represented on iterated sagittal slices of an MNI normalized brain template. (**e**) Volumetric overlap of ALE clusters with HCP parcellation territories.

**Figure 4 jpm-13-01384-f004:**
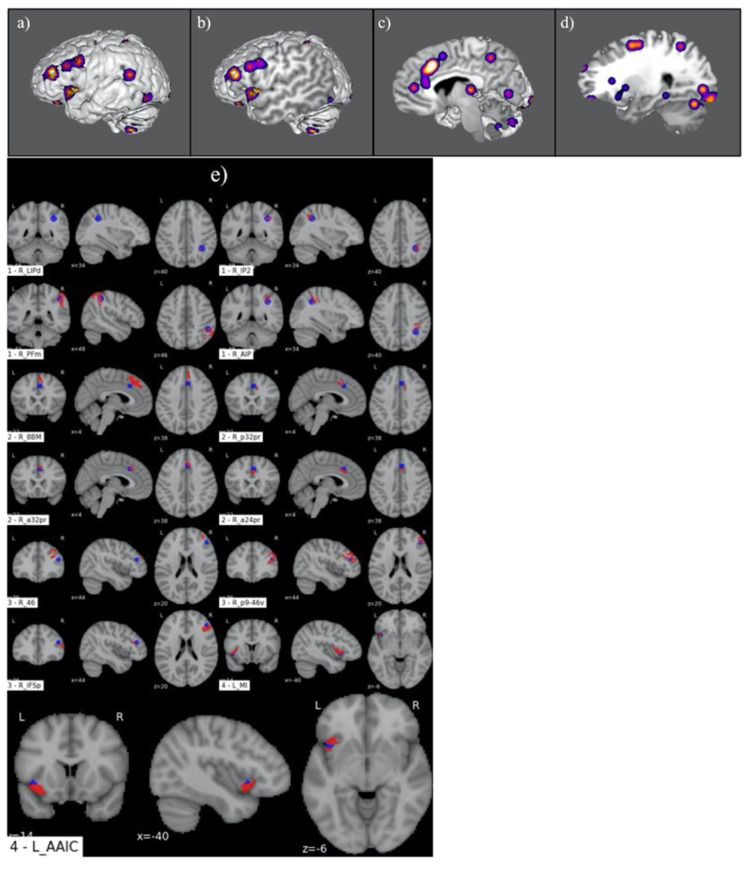
Graphical representation of ALE for judgement under conditions of risk. (**a**–**d**) Illustration of ALE clusters represented on iterated sagittal slices of an MNI normalized brain template. (**e**) Volumetric overlap of ALE clusters with HCP parcellation territories.

**Figure 5 jpm-13-01384-f005:**
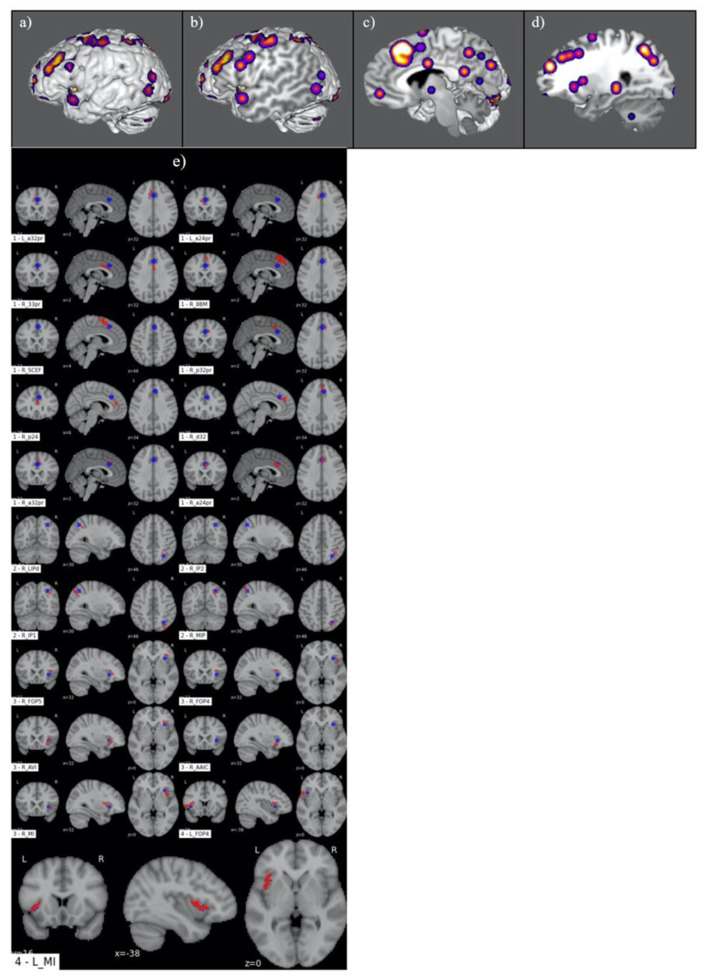
Graphical representation of ALE for interpersonal judgement. (**a**–**d**) Illustration of ALE clusters represented on iterated sagittal slices of an MNI normalized brain template. (**e**) Volumetric overlap of ALE clusters with HCP parcellation territories.

**Figure 6 jpm-13-01384-f006:**
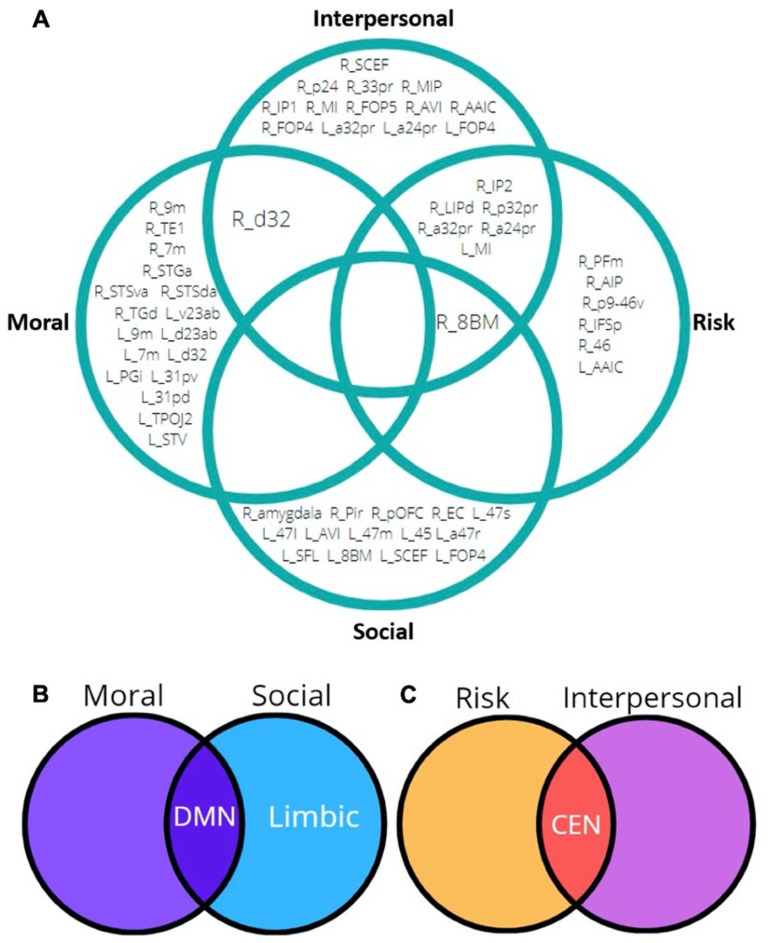
(**A**) Summary of HCP parcellation co-activation across ALE sub-analyses. Note that area L-FOP4 was activated across both social and interpersonal judgement. (**B**,**C**) Summary of network co-activation across ALE sub-analyses. Also note social judgement uniquely activated the right-sided limbic network.

## Data Availability

Data are available upon request.

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
