# Peer review of "Parcellation-Based Connectivity Model of the Judgement Core"

_jpm, 2023, doi:10.3390/jpm13091384_

Round 1
Reviewer 1 Report
1. Abstract was too long to read.
2. In what kind of activity we need the judgement. Please explain briefly.
3. complex thinking and behavior will change from person to person?
4.Why did you conduct your studies with healthy participants alone?
5. Did you conduct any studies with stressed person?
6.Improve your literature survey
7. Need more explanation for experimental setup
8. Conclusion was not sufficient for this study
Author Response
Please see attached response to reviewer form.

Reviewer 2 Report
This article performed a coordinated-based meta analysis of existing published reports which technique has been used before. However, there are also reports combining the CBMA analysis with a behavioral analysis to get more insight about the relationship between brain and behavior. In order to improve the manuscript I recommend that a behavioral analysis be added to this study in order to further understand the relationship between judgement core and brain function. If possible, I would like to see the correlation between brain fMRI activation and judgement behavioral scores.
Author Response

(The authors gave the same response as above.)
